# Encapsulation Reduces the Deleterious Effects of Salicylic Acid Treatments on Root Growth and Gravitropic Response

**DOI:** 10.3390/ijms232214019

**Published:** 2022-11-14

**Authors:** Jimmy Sampedro-Guerrero, Vicente Vives-Peris, Aurelio Gomez-Cadenas, Carolina Clausell-Terol

**Affiliations:** 1Departamento de Biología, Bioquímica y Ciencias Naturales, Universitat Jaume I, 12071 Castellón de la Plana, Spain; 2Departamento de Ingeniería Química, Instituto Universitario de Tecnología Cerámica, Universitat Jaume I, 12071 Castellón de la Plana, Spain

**Keywords:** DR5::GFP auxin sensor, encapsulation process, indole acetic acid

## Abstract

The role of salicylic acid (SA) on plant responses to biotic and abiotic stresses is well documented. However, the mechanism by which exogenous SA protects plants and its interactions with other phytohormones remains elusive. SA effect, both free and encapsulated (using silica and chitosan capsules), on *Arabidopsis thaliana* development was studied. The effect of SA on roots and rosettes was analysed, determining plant morphological characteristics and hormone endogenous levels. Free SA treatment affected length, growth rate, gravitropic response of roots and rosette size in a dose-dependent manner. This damage was due to the increase of root endogenous SA concentration that led to a reduction in auxin levels. The encapsulation process reduced the deleterious effects of free SA on root and rosette growth and in the gravitropic response. Encapsulation allowed for a controlled release of the SA, reducing the amount of hormone available and the uptake by the plant, mitigating the deleterious effects of the free SA treatment. Although both capsules are suitable as SA carrier matrices, slightly better results were found with chitosan. Encapsulation appears as an attractive technology to deliver phytohormones when crops are cultivated under adverse conditions. Moreover, it can be a good tool to perform basic experiments on phytohormone interactions.

## 1. Introduction

Physiological processes that allow the correct development of plants are controlled by biomolecules [1]. Among them, plant growth regulators (PGRs) are active substances that can have a natural origin, such as the phytohormones, but also can be chemically synthesized. The phytohormones, which are low molecular weight organic compounds of endogenous origin [2], are distributed in different plant tissues and, in small quantities, are capable of modulating morphogenetic and physiological processes [3]. Moreover, phytohormone effects are complex because their action, in some cases, is indirect and the signal can initiate in a plant tissue far from where the final effect is observed [4]. It is important to determine the role of phytohormones and their interactions since PGRs are commonly used to modify developmental patterns and growth rates in seeds, shoots and roots [5], with high economic and agronomic benefits. Climate change is enhancing the incidence and intensity of abiotic stresses in the fields, giving rise to a complicate scenario where crops are gradually reducing yields and fruit qualities [6,7]. PGRs have a remarkable potential to induce plant responses to stress, contributing to the adaptation of crops to adverse environments [8].

Salicylic acid (SA), fully recognized as a PGR in the 1990s [9], has an aromatic benzene ring in its structure and belongs to the family of phenolic compounds [10]. SA participates in key biological processes such as plant development, antioxidant system, nitrogen metabolism and photosynthesis regulation, among others [11]. In addition, it is an important component of plant tolerance to biotic stresses [12,13,14]. SA has an important role in plant responses to abiotic stresses in cooperation with other phytohormones such as jasmonic acid (JA) and abscisic acid (ABA), in a controlled cross-talk that allows the plant to adapt to drought, highlights, high temperatures and high salinity [15,16]. The capacity of exogenous SA for inducing stress tolerance mechanisms has been deeply studied. In fact, different application methods, such as the addition to the nutrient solution or irrigation water and spraying or soaking the seeds have been tested [17,18]. A good example is the protective effect of SA against salt stress, where the addition of SA is capable of dispelling the toxicity symptoms in many plant species, restoring membrane potential and improving photosynthetic capacity and antioxidant protection [19,20]. When considering an SA treatment, several aspects must be taken into account, such as the SA concentration, the plant species, the age of the plant and the duration of the treatment [21]. Indeed, choosing the optimal dose of SA is an important point to consider because a small amount will be rapidly absorbed by the plant, decreasing its protective effect over time and, on the contrary, a large amount will cause stress in the plant [22].

Encapsulation is an efficient solution to control the release of SA and, therefore, the applied concentration. Encapsulation is a relatively new technology where an active agent is loaded into a carrier matrix [23] of a different nature, often polymer-based. The benefits of this process are many, such as (a) active agent protection during the storage and the application process, (b) decrease in the amount of active agent required, and (c) controlled release of the encapsulated molecule [24,25]. The encapsulated active agents can be small or large molecules, such as proteins, drugs or dyes, and capsules, which are generally organic or inorganic polymers, fatty acids or lipids [26]. In the last years, various polymeric shells had been developed, for example: polyuria, polysaccharides such as chitosan and alginate, aliphatic polyesters and amorphous silica [27], which are efficient for formulating capsule-active agent mixtures, which requires an aqueous system without changing room temperature [28]. Authors have recently shown that chitosan and amorphous silica are effective carriers of plant-derived substances [29], immobilizing and releasing them in a controlled manner [30].

The aim of this work was to study the differences among treatments with free SA and encapsulated SA (in silica or chitosan) in *Arabidopsis thaliana* plants. The effect of SA concentration was studied, demonstrating that free and encapsulated SA affect plants in different ways. The encapsulation of SA using both capsules proved to be an effective method to reduce the negative effects of the phytohormone accumulated in roots and rosettes through its controlled release and its correct delivery to the plant. The results provide a desirable encapsulated product that can be used in agriculture to mitigate the adverse effects of different stresses on plants.

## 2. Results and Discussion

### 2.1. Encapsulation Decreases the SA Negative Effect on Primary Root Growth

Phytohormones are essential molecules that control the correct development of plants through a regulated homeostasis, which may be disrupted by the exogenous application of PGRs [31]. SA works as a plant defense activator and a growth regulator. However, applications of SA to concentrations greater than 1 mM inhibit seed germination and plant growth [32]. To control the release of SA and reduce its negative effect, encapsulated samples were developed in a previous study [29]. In this study, we found that the process differs between capsules. In the case of amorphous silica, an SA gradient from the surface to the centre of the encapsulated sample was observed, with a potential saturation of the silica porous surface depending on the Si:SA ratio. In the case of chitosan, this process is produced by entrapment of the bioactive molecule by the chitosan polymeric chains and subsequent cross-linking with TPP-Na. These differences in the encapsulation process were reflected in the SA release rates being faster in the case of silica. However, no differences were observed in the release method, produced in both cases by breakage of the carrier matrix structure and not by diffusion of the encapsulated molecule. The samples selected for their best results inhibiting fungal growth were those with the lowest ratio tested with both capsules: 1:0.5 for Ch:SA and 1:0.25 for Si:SA (Figure 1). In this work, doses of SA (either free or encapsulated), ranging from 1 to 500 µM, were tested to measure their effect on Arabidopsis root growth (Figure 2). Data indicate that after 5 days of treatment, a significant decrease in the length of the primary root was observed in free-SA-treated plants in all the doses tested (Figure 2b–f). Plants treated with SA encapsulated with Si had a root length similar to that of the control plants at doses of 1 and 10 µM (Figure 2g–k) and those treated with Ch:SA at doses of 1, 10 and 50 µM (Figure 2l–p). Intermediate lengths (shorter that controls but longer than free SA treated plants) were found at higher doses for both capsules.

The growth rate of the Arabidopsis roots was calculated for each treatment and dose, to determine a value that depends on the root tip elongation [33]. The growth rate in the roots of the control plants was approximately 0.7 cm on the 1st day and 0.4 cm on the rest (2nd to 5th day) (Figure 3 and Appendix A). On the first day of analysis, the root growth rate in plants treated with free SA was similar to that of the controls only at the 1 µM dose. However, from the 2nd day and until the end of the test, the growth rate was considerably reduced (Figure 3a). The final root length was 1.49 cm when treated with SA at a 1 µM dose (compared to the 2.39 cm of the control roots), and much shorter in those plants treated with higher concentrations of free SA. In other plant systems, constantly-applied free SA had a strong effect on primary root growth, due to inhibition of cell elongation at low doses (≤100 µM) [34] and complete stoppage at high doses (≥100 µM), as occurs in other dicots such as bean [35] and cucumber [36]. Interestingly, plants treated with Si:SA and Ch:SA at the lowest doses had a root growth rate similar to the control plants (Figure 3b,c). When the three treatments were compared (free SA, Si:SA and Ch:SA) throughout the test period, it was observed that, in general, the encapsulation process (regardless of the capsule used) reduced the adverse effect of SA on root growth (Figure 2 and Figure 3 and Appendix A).

The difference in root length among plants treated with the different products could be due to the progressive release of SA from the capsules [37], decreasing the amount of SA in the medium (available then to the plant). When treating with free SA, the amount of molecule available to the plant in the first moments coincides with the total dose, quickly stressing the plant and causing the death of root cells [38]. In this regard, results of the root growth test performed from 0 to 12 h (Appendix A) corroborate the results shown in Figure 3. As can be noticed, the least toxic treatment was again Ch:SA, compared to Si:SA and free SA. The experiment confirmed that free SA becomes toxic very rapidly in the plant. No effects on growth rate were observed when plants were treated with empty Si or Ch capsules (Appendix A).

### 2.2. The Encapsulation of SA Allows for the Correct Development of the Rosette

The effect of SA on rosette size is an important parameter of healthy plants. The rosette area in early stages is proportional to biomass [39]. In fact, results show that SA treatment affects the rosette size in a dose-dependent manner, as rosettes are smaller as free SA concentration increases (Figure 4 and Appendix A). The constitutive response to high concentrations of SA may cause morphological alterations such as dwarfism [40]. As shown in the Appendix A, and quantified in Figure 4a, rosettes of plants treated with Si:SA showed a size similar to those of the control at concentrations of 1 and 10 µM. However, size decreased as the concentration of encapsulated SA increased (50, 100 and 500 µM). Ch:SA treatment was even better as treated plant rosettes were as large as the controls at 1, 10 and 50 µM doses. Comparison among treatments reveals that rosettes treated with free SA had a very small size, even at the lowest doses (area was approximately half of that of the control plants, Figure 4b–f). However, encapsulation (regardless of the matrix used) decreases the adverse effects on rosettes.

Rosette and root results are consistent since high concentrations of SA available in the growth medium increase the SA absorbed by the plant [41], which is possibly transported to the rosette and accumulated in the aerial tissues until plants are intoxicated. No effects on the rosette size were observed when plants were treated with empty Si or Ch capsules (Appendix A).

### 2.3. The Encapsulation of SA Prevents Alteration of Root Gravitropism

Roots are able to feel and respond to gravity changes, always growing in the direction of the gravity vector, in what is referred to as positive gravitropism [42,43]. As depicted in Figure 5 and Appendix A, plants treated with free SA at the lowest doses showed a similar root gravitropism to the control plants, 8 h after the plant orientation change. However, roots lost the capacity of reorienting when plants were treated with free SA at 50 or 100 µM. At the highest dose (500 µM), roots became mostly agravitropic (unable to turn). As shown in Appendix A, the same general conclusion can be drawn 24 h after the plant rotation. Previous reports showed that exogenous SA controls the root change orientation in an IAA crosstalk network [44] and reduces the root orientation angle in a dose-dependent manner, proving that SA has a negative effect on gravitropism [45,46]. However, the altered gravitropism response was significantly reverted when treating plants with the encapsulated hormone. Plants treated with SA encapsulated with any of the two capsules showed gravitropic roots (Figure 5 and Appendix A). After 24 h of changing angle orientation, plants treated with the 1, 10 and 50 µM doses of Si:SA had a root orientation identical to the controls (Appendix A). Treatments with Ch:SA were still less aggressive regarding to this parameter and only plants treated with the highest dose had problems changing their angle orientation (Appendix A). These data show that exogenous SA alters the root gravitropism response in Arabidopsis plants until it is abolished at high doses. We hypothesise that this treatment induces changes in gene expression, especially in those genes related to auxin synthesis and transport, which subsequently alter root patterning and growth direction [47]. However, encapsulation decreases SA’s deleterious effects on root gravitropism by delaying the presence of SA in the medium. No effects on the gravitropism response were observed when plants were treated with empty Si or Ch capsules (Appendix A).

### 2.4. The Encapsulation of SA Modulates Endogenous SA Accumulation in Plants

A profile of phytohormones SA, JA, ABA and IAA was obtained after 28 days of treatment, both from the roots and the rosettes. The most affected plants, with small tidied up rosettes and short and agravitropic roots, were those treated with free SA at 100 and 500 µM doses (Figure 6e–f). These results agree with Section 3.1, Section 3.2 and Section 3.3, where the increase in SA concentration impaired plant growth. The negative effect of SA was reduced by the encapsulation process as reported before (Figure 2, Figure 4 and Figure 5). These results confirm that the capsules have an optimal design as carriers and shields of SA, as well as for their gradual release, controlling the potential toxicity of the phytohormone [48]. Plants had higher levels of endogenous SA both in the rosettes (Figure 7) and the roots (Figure 8) after the treatment with free SA. However, treatments with the encapsulated SA importantly reduced these increased levels of endogenous SA. In detail, significant increases in the endogenous SA in rosettes was obtained after treating plants with all doses of free SA and only at the doses of 50, 100 and 500 µM for treatments with both encapsulated samples (Figure 7).

Endogenous SA levels in the roots of the treated plants followed a similar pattern, and the roots treated with free SA had the highest values of endogenous SA, followed by those treated with Si:SA and Ch:SA, all of them at the doses of 1, 10, 50 and 100 µM (Figure 8b–e). At the 500 µM dose, no differences in root SA concentrations were detected among plants under the different treatments (Figure 8f). However, it is important to highlight that plants grew differently depending on the treatment (Figure 6f,k,p). Hormones are important signals involved in the regulation of the cell division and size in plants [49,50].

Therefore, we propose that exposition to free SA increases endogenous hormone levels from the beginning of the experiment and this early increase causes short roots and small and pale rosettes. Encapsulation is able to control the early uptake of SA, limiting the amounts of hormone that the roots absorb and translocate to the rosette. This regulation of the endogenous SA levels reduces the damaging effect of the treatments and, therefore, has less effect on plant phenotypes.

### 2.5. The Encapsulation of SA Modulates Endogenous IAA Accumulation in Roots

The IAA is the main auxin that regulates root elongation and several developmental processes in plants, such as tissue differentiation, cell division, response to different pathogens, etc. [51,52]. Endogenous levels of IAA in the roots showed a noticeable depletion with the increase in the dose of free SA, reaching barely detectable values in plants treated with the 500 µM dose (Figure 9a). However, in plants treated with the encapsulated SA samples, fluctuant values were observed compared to the control plants (Figure 9b–f). In fact, SA regulates root growth together with IAA in a balanced pathway, which could be altered by the change in the levels of any of them (in this case the SA levels) and which modifies the root response [53], as roots are highly sensitive to fluctuations of IAA levels [54]. To confirm the effect of SA treatments on IAA levels, a reporter line of *Arabidopsis thaliana* “DR5::GFP” was used [55]. The DR5 system allows for monitoring auxin levels, especially in the root tip, since this group of cells show important an accumulation of IAA in response to any stimulus [56]. In general, IAA levels in roots (Figure 9) correspond with the activity of the DR5::GFP auxin sensor in the quiescent centre (Figure 10), although some differences are observed, probably due to the specific cells monitored in the DR5 system (the quiescent centre) versus the bulk levels in the whole primary root detected in the analytical assay.

As depicted in Figure 10b–d, a progressive increase in the activity of DR5:GFP in plants treated with 1, 10 and 50 µM free SA doses was observed. Interestingly, in plants treated with Si:SA and Ch:SA, fluorescence still increased at the 100 µM dose (Figure 10j,o) but had a slight decrease at the 500 µM dose (Figure 10k,p). However, in plants treated with free SA, a fluorescence decrease was observed at the 100 µM dose (Figure 10e) and no activity could be found at the 500 µM dose (Figure 10f). This may be due to the fact that high doses of SA (≥100 µM) reduce auxin levels, decreasing the activity of the DR5::GFP auxin sensor in the quiescent centre [57]. Therefore, it seems that an effect of SA can be the suppression of the auxin flow from the stem to the root tip. This relationship between SA and IAA is related to their functions, and is well described in the literature, suggesting that the gradient of IAA plays an essential role in the correct dynamics of root growth [58]. In fact, the encapsulated samples prevented IAA levels from declining further at doses of 100 and 500 µM, allowing roots to grow in contrast with roots treated with free SA. The different treatments with SA cause small changes in the levels of JA and ABA in a random way that did not allow for identifying any pattern of regulation (Appendix A). From these data, it can be concluded that one of the improvements of the use of encapsulated SA is to prevent auxin flow interferences.

### 2.6. General Comparison of Free SA vs. Encapsulated SA Treatments

Results of root growth, rosette area, root angle and phytohormone levels (SA, JA, ABA, IAA) from the roots were evaluated in a Principal Component Analysis (PCA) plot to reduce the dimensionality of our datasets and avoid losing important information [59]. To establish the relationship among the variables analysed in the treatments, a plot of variables was made to correlate the importance of each variable in the main component [60]. According to the results shown in the Variables-PCA plot (Appendix A), there is a positive correlation and a good quality representation of the variable in the principal component (cos2) among root growth per day, root angle at 8 h and SA content, grouped in the same direction, and with a less cos2 and root growth per hour. The IAA content had a positive correlation as well. However, a negative correlation is observed among the rosette area, JA content and ABA content, since these variables are represented on the opposite side to the origin and/or in the opposite quadrants (Appendix A). According to these results, SA and IAA levels explain the direct relationship between SA levels and the affection in the roots of the treated plants. Therefore, the hypothesis of exogenous SA treatments modifying endogenous IAA levels is further supported, allowing us to conclude that part of the deleterious effect of exogenous SA is due to the reduction of root IAA levels and that encapsulation avoids this process. In the same way, a correlation matrix was made to highlight the importance of the variables in the two main components. The results matrix showed that all variables were found in the first principal component (Dim1), except that of JA content, which is found in the second principal component (Dim2) (Appendix A). This graph shows that the variables are perfectly represented by only two main components, in this case, Dim.1 and Dim.2 [61].

The PCA is also capable of evaluating the correlation among several treatments, growth parameters and internal phytohormone levels in plants [62]. The individual PCA of the treatments and the different doses of SA revealed that the two main components covered approximately 81.8% of the total variance (66.6% and 15.2% for Dim1 and Dim2, respectively) (Figure 11). In the first Dim1, the treatments with free SA are grouped, and a large separation between the doses of free SA and Si:SA at 100 and 500 µM doses are observed, so free SA and encapsulated SA (at these doses) caused important morphological changes in plants. Indeed, these negative effects become stronger when the SA dose increases, raising the point distribution variability of Free SA and Si:SA treatments at the highest doses values in the PCA (Figure 11). On the other hand, in the Dim2, the treatment with Si:SA at doses of 1, 10 and 50 µM and those of Ch:SA at all doses are grouped, which indicates that plants were not seriously affected (as these values are not different from those of the control with a similar profile and less variability), placing them within a range in which the plant is affected but still is able to develop.

## 3. Materials and Methods

After an initial study of the effect of SA doses on the development of Arabidopsis plants, the following experiments were performed: First, the effect of the SA treatments on root and rosette physiology and morphology was evaluated both at short and long periods. Then, the gravitropic response of treated plants was studied and, finally, DR5::GFP lines were used to evaluate the auxin fluxes in roots. Figure 1 shows a summary of the methodology used in this work: (a) formulation of treatments and Arabidopsis seed sown, (b) analysis of SA effect on root and rosette growth and root gravitropism, (c) extraction and quantification of phytohormones, (d) visualization of the auxin-specific reporter gene DR5 in roots and (e) statistical and PCA analyses.

### 3.1. Materials and Plant Growth Conditions

Salicylic acid (SA), pyrogenic amorphous silica HDK^®^ S13 (Si) and chitosan DG CHI 0.20 g/mL and 85% deacetylated (Ch) were purchased from Sigma-Aldrich (St. Louis, MO, USA), AOXIN (Shanghai, China) and WACKER (Barcelona, Spain), respectively. *Arabidopsis thaliana* wild-type (Col-0) seeds were obtained from the Nottingham Arabidopsis Stock Centre (Nottingham, UK), and *Arabidopsis thaliana* DR5::GFP line was obtained from the Arabidopsis Biological Resource Center (Columbus, OH, USA). Seeds were surface sterilized with 1% *v*/*v* sodium hypochlorite and 0.01% *v*/*v* Tween 20 solution for 10 min with moderate incubation, washed in triplicate with sterile distilled water, and sown in 9 × 15 cm petri dishes containing Murashige and Skoog medium 0.5% (Duchefa, Haarlem, The Netherlands), sucrose 1% (Merck Millipore, Darmstadt, Germany) and European Bacteriological Agar (Condalab, Madrid, Spain). Seeds were germinated under no stress conditions and petri dishes were vertically arranged (Figure 1) in growth chambers (SANYO MLR-350, Sakata, Gunma, Japan) for 5 days under 16 h light/8 h dark cycles at 22.5 °C and 60% relative humidity. After this period, plants were transferred to the media containing the different SA treatments (see Section 3.2.) and kept in the same growing conditions for different periods.

### 3.2. SA Treatment Conditions

The following concentrations of SA were used for treatments: 1, 10, 50, 100 and 500 μM. For each SA concentration, in addition to non-encapsulated SA (referred to as free SA), 1:0.25 ratio of Si:SA and 1:0.5 ratio of Ch:SA encapsulated samples were obtained by spray-drying the aqueous suspensions prepared by planetary mixing (Pulverisette^®^, Fritsch, Idar-Oberstein, Germany) of the specific amounts of SA with silica and chitosan, respectively, following the experimental procedure detailed in a previous publication [29]. In brief, the Si:SA sample was prepared by mixing the respective amount of SA with 320 mL distilled water (15 min at 120 rpm), adding the amorphous silica stepwise and homogenizing the mixture (1 h at 180 rpm). Ch:SA was prepared by mixing 138.6 mL of distilled water and 1.4 mL of acetic acid (5 min at 150 rpm) and by adding, in successive steps, 4.2 g of chitosan (15 min at 210 rpm), 1.4 mL of tween 80 (15 min at 210 rpm), the appropriate amount of SA pre-dissolved in dichloromethane (15 min at 210 rpm) and 2.1 g of TPP-Na pre-dissolved in 137.9 mL of distilled water (1 h at 210 rpm). Spray drying was performed in a SD-06 spray drier (LabPlant, Filey, UK), with a standard 0.5 mm nozzle and the following standard conditions: inlet temperature 150 °C, spray flow 10 mL/min, drying air fan 80% and compressed air pressure 1.5 bar. Encapsulated samples and free SA were mixed with the culture medium and poured in petri dishes.

### 3.3. Determination of Root Growth, Rosette Area and Root Gravitropism

After transferring plants to the different treatments, petri dishes were scanned with an Epson perfection v600 photo scanner, and root length measured by Image J 1.53t software using the obtained images (Figure 1b1). Dishes were scanned each hour for 24 h and each day up to 5 days, calculating root length from the images. Five-day growth plants were removed from the dish, and their rosettes (separated from their roots) were scanned. Finally, for the gravitropism test, dishes were tilted 90°. Every hour (from 0 h to 24 h), the dishes were scanned, maintaining the same inclination. Root angle changes were measured by the same Image J software (Figure 1b2).

### 3.4. Extraction and Phytohormones Analysis

After transferring plants to the different treatments, plants were grown for 4 weeks in 90° tilted dishes. Then, plants were sampled and both the rosettes and the roots were frozen with liquid nitrogen and stored until further analysis. Extraction and analysis were carried out as described in [63] with few modifications. Briefly, 0.2 g of plant tissue were extracted with 1 mL of acetonitrile 50% in a ball mill (Millmix20, Domel Železniki, Slovenia) after spiking with 2.5 ng of [^2^H_5_]-indole acetic acid (IAA) and 25 ng of the following molecules: [^13^C_6_]-SA, dehydro jasmonic acid (DHJA) and [^2^H_6_]-ABA. Extracted samples were sonicated and centrifuged to remove debris. Then, 1 mL of the sample was charged in an “Oasis HLB 1 cc Vac Cartridge, 30 mg (Waters, Mildford, CT, USA)” column with 500 μL of acetonitrile 30%, collecting the eluent. Phytohormones SA, JA, ABA and IAA were determined in rosettes and roots by high performance liquid chromatography coupled online with a triple quadrupole mass spectrometer (Micromass, Manchester, UK) through an orthogonal Z-spray electrospray ion source [64] (Figure 1c).

### 3.5. Fluorescence Analysis

The DR5::GFP sensor system has been widely used to study the auxin response because it contains regulatory elements suitable for inferences about auxin levels. Five-day-old *Arabidopsis thaliana* DR5::GFP plants were transferred to different treatments. After 5 days, whole plants were taken from the medium, placed on microscope slides and visualized under the microscope with a 40× objective. Fluorescence images were acquired by a “Nikon Eclipse 80i fluorescent microscope (MicroscopyU, Melville, NY, USA)”, equipped with an epifluorescence GFP filter (Figure 1d). Fluorescence intensity and exposure time were at 30% and 200 ms, respectively. Images were treated with plugin FIJI from Image J software.

### 3.6. Statistical Analysis and Principal Component Analysis (PCA)

Treatments consisted of three replications and, at least, ten plants for each replication. SPSS version 21 software was used for statistical analysis and one-way analysis of variance test (Anova) with Bonferroni correction to determine significant differences between treatment groups at *p* ≤ 0.05. Correlation matrix graphic and Individual—PCA/Variables—PCA were constructed using R package corrplot [65] and R package factoextra [66]—R package FactoMineR [67], respectively (Figure 1e).

## 4. Conclusions

The experiments performed demonstrate that encapsulation prevents the uncontrolled release of SA and decreases the adverse pleotropic effects of the free SA treatment on plant physiology. Plants are able to take up free SA when it is available in the medium. This rapid uptake affects the structure and length of the roots, and the size and architecture of the rosettes in a concentration-dependent manner, because of both the high SA accumulation in both tissues and the decrease in the IAA accumulation and activity, especially in the root. However, changes in IAA levels in root germ cells, due to fast uptake of SA, can be prevented by its encapsulation, reducing the amount available for the plants. SA encapsulation with silica or chitosan results in a controlled release of SA and, therefore, in fewer negative effects (when compared with free SA), considering that plants suffer impaired physiological and morphological responses. Encapsulated samples at the lowest doses have no impact on treated plants, with Ch:SA being the least harmful. At the highest doses (100 and 500 µM), plants are more damaged because the amount of SA is excessive. Differences among treatments are consistent with the PCA, showing that encapsulation is a useful method to control deleterious SA effects. When comparing capsules, Ch has a lower impact on treated plants, which maintain a relatively normal development. Therefore, the highest doses can be used in comparison with the Si capsule. Future work will be aimed at studying the effect of encapsulated hormones on plants under conditions that emulate the climate change to evaluate the positive effect that encapsulated samples can have on plant tolerance to biotic and abiotic stress conditions. In these future studies, the system developed in this work with Arabidopsis growing in controlled conditions can be a key tool to decipher the optimal carrier matrix, hormone dose range and optimal plant growth conditions before extrapolating the experimental design to greenhouses or fields.

## Figures and Tables

**Figure 1 ijms-23-14019-f001:**
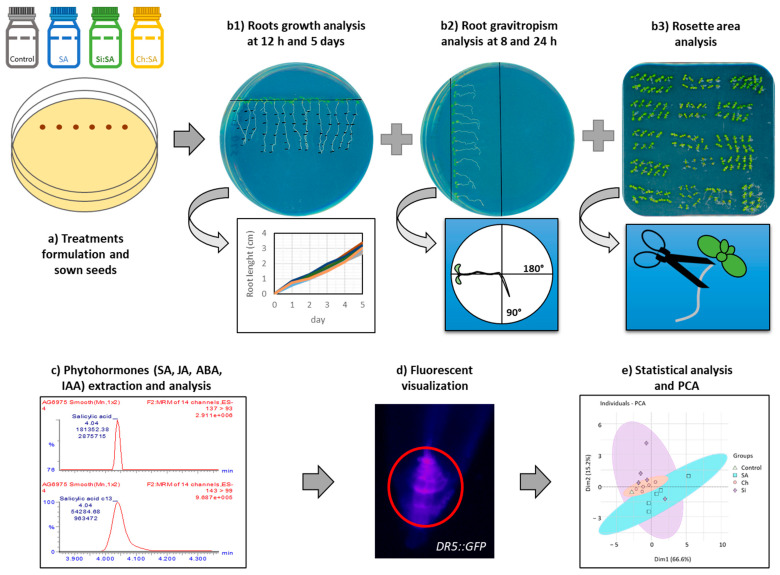
Experimental method developed to determine the effect of free and encapsulated SA (Si:SA and Ch:SA) in *Arabidopsis thaliana* plants.

**Figure 2 ijms-23-14019-f002:**
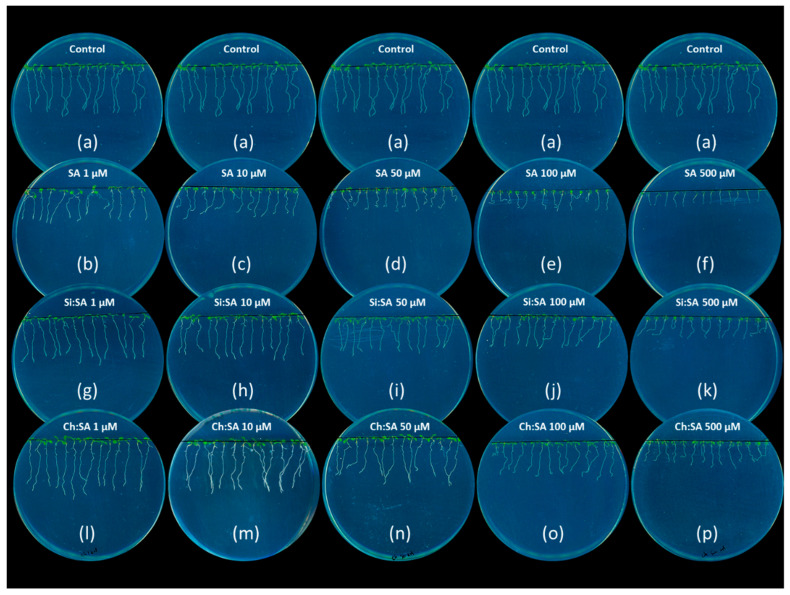
Effect of free SA, Si:SA and Ch:SA on root growth in Col-0 Arabidopsis plants. Five-day-old plants were transferred to media containing the different SA treatments and pictures were taken 5 days later.

**Figure 3 ijms-23-14019-f003:**
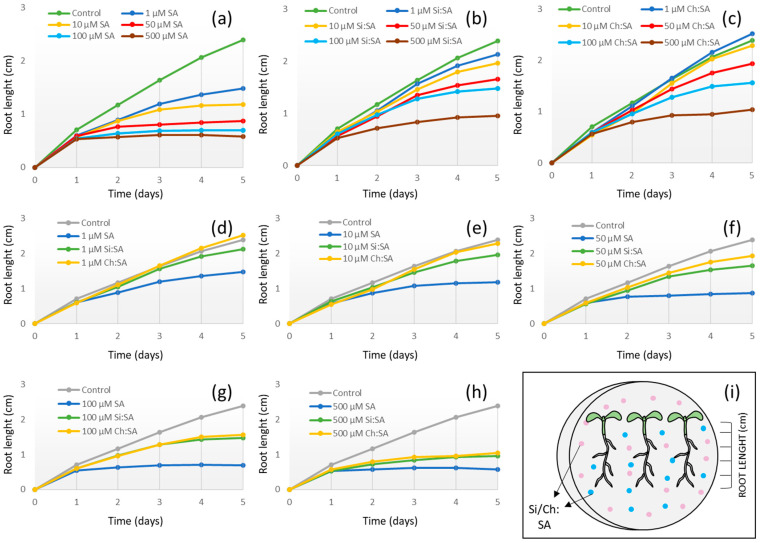
Effect of free SA, Si:SA and Ch:SA on root growth in Col-0 Arabidopsis plants. Five-day-old plants were transferred to media containing the different SA treatments and root length was measured daily (Scheme (**i**)). Graphs (**a**–**c**) compare root length among the doses at each treatment, and graphs (**d**–**h**) compare root length among the treatments at each dose.

**Figure 4 ijms-23-14019-f004:**
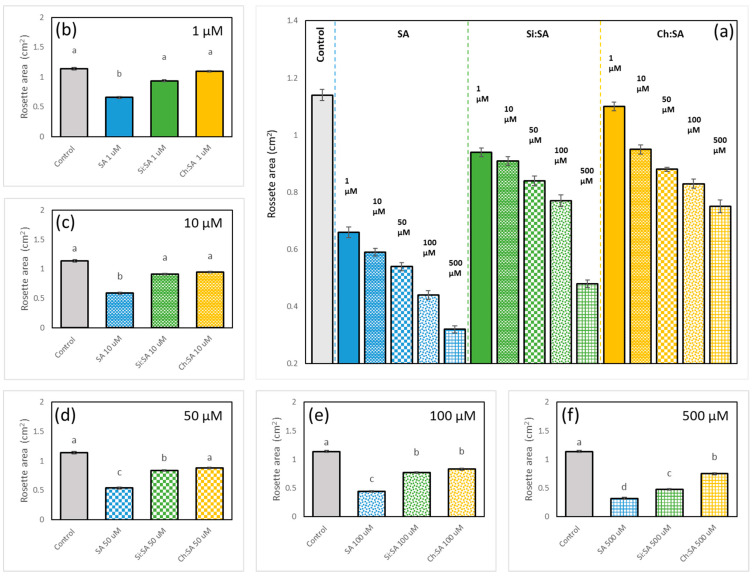
Effect of free SA, Si:SA and Ch:SA on the rosette size in Col-0 Arabidopsis plants. Five-day-old plants were transferred to media containing the different SA treatments and size was measured 5 days later. Graph (**a**) depicts the rosette area for the three treatments and at all doses, and graphs (**b**–**f**) compare the rosette area among the treatments at each dose (1 to 500 µM, respectively). Different letters indicate significant differences among treatment groups at *p* ≤ 0.05.

**Figure 5 ijms-23-14019-f005:**
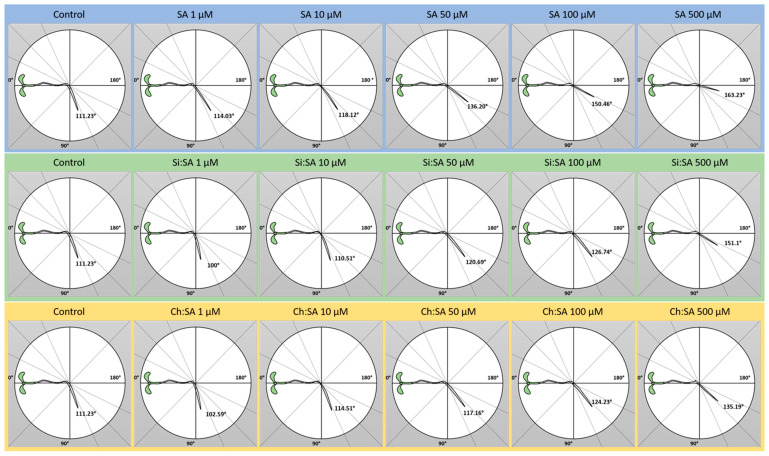
Effect of free SA, Si:SA and Ch:SA on root reorientation in Col-0 Arabidopsis plants. Five-day-old plants were transferred to media containing the different SA treatments and the root angle was measured 8 h later.

**Figure 6 ijms-23-14019-f006:**
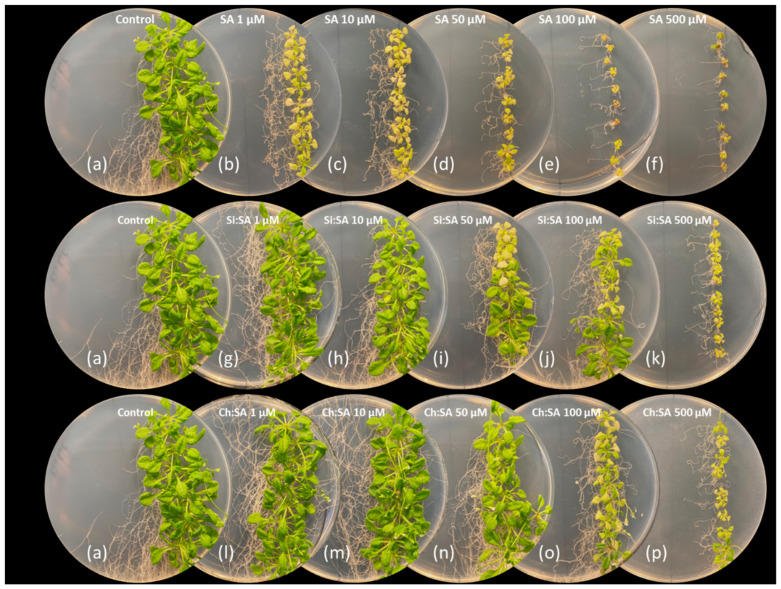
Effect of free SA, Si:SA and Ch:SA on plant performance in Col-0 Arabidopsis plants. Five-day-old plants were transferred to media containing the different SA treatments and pictures were taken 28 days later.

**Figure 7 ijms-23-14019-f007:**
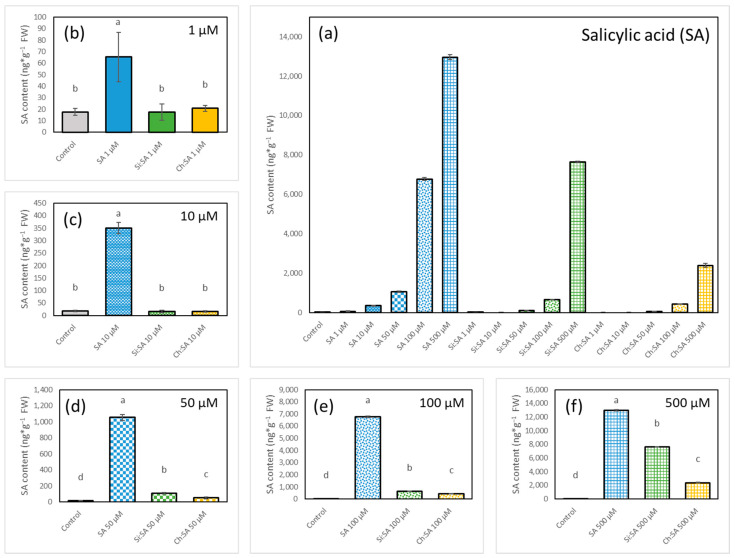
Effect of free SA, Si:SA and Ch:SA on endogenous SA levels in the rosettes of Col-0 Arabidopsis plants. Five-day-old plants were transferred to media containing the different SA treatments and plant hormones were measured 28 days later. Graph (**a**) depicts SA levels in the three treatments at all doses, and graphs (**b**–**f**) compare SA levels among the treatments at each dose. Different letters indicate significant differences among treatment groups at *p* ≤ 0.05.

**Figure 8 ijms-23-14019-f008:**
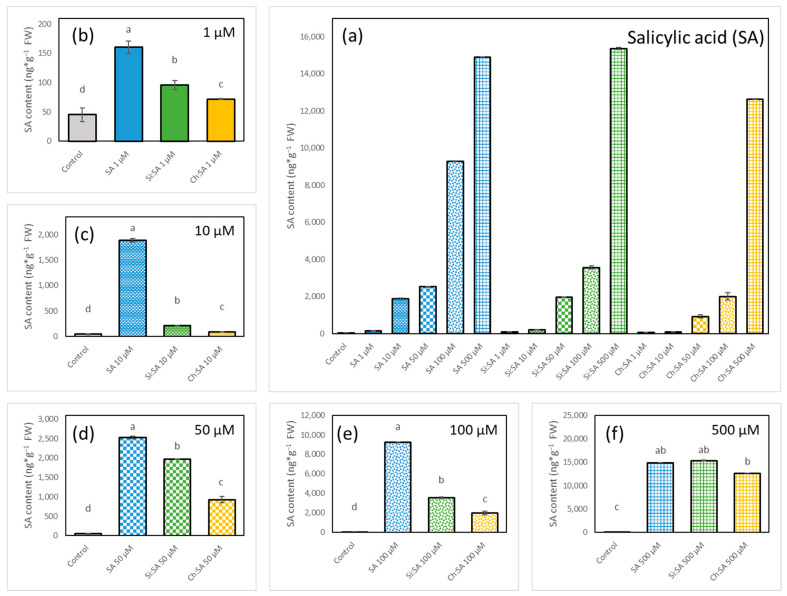
Effect of free SA, Si:SA and Ch:SA on endogenous SA levels in the roots of Col-0 Arabidopsis plants. Five-day-old plants were transferred to media containing different SA treatments and plant hormones were measured 28 days later. Graph (**a**) depicts SA levels in the three treatments at all doses, and graphs (**b**–**f**) compare SA levels among the treatments at each dose. Different letters indicate significant differences among treatment groups at *p* ≤ 0.05.

**Figure 9 ijms-23-14019-f009:**
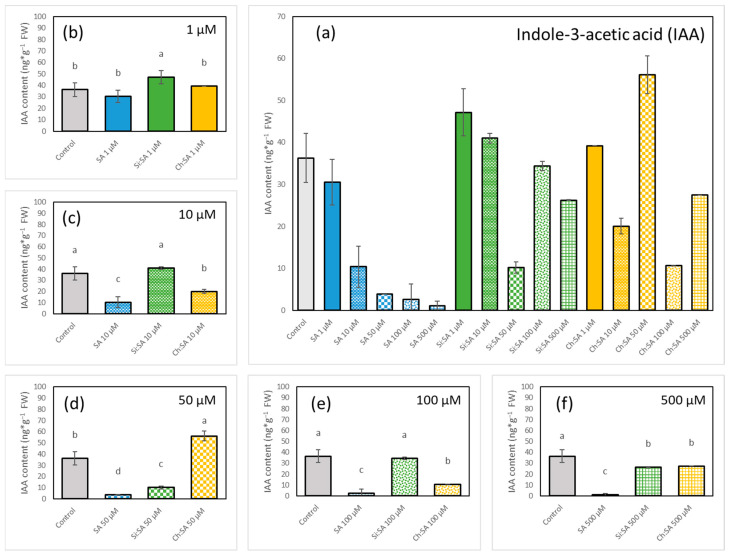
Effect of free SA, Si:SA and Ch:SA on endogenous IAA levels in the roots of Col-0 Arabidopsis plants. Five-day-old plants were transferred to media containing the different SA treatments and plant hormones were measured 28 days later. Graph (**a**) depicts IAA levels in the three treatments at all doses, and graphs (**b**–**f**) compare IAA levels among the treatments at each dose. Different letters indicate significant differences among treatment groups at *p* ≤ 0.05.

**Figure 10 ijms-23-14019-f010:**
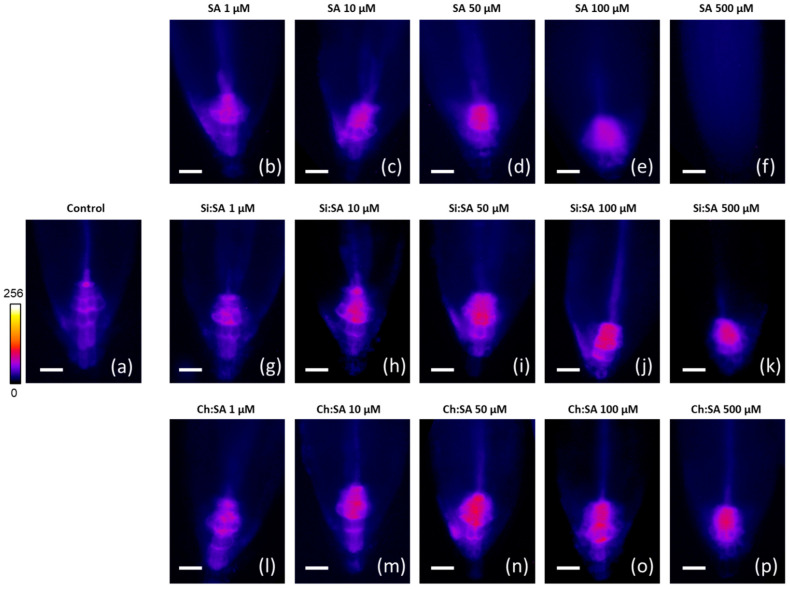
Effect of free SA, Si:SA and Ch:SA on root fluorescence in Arabidopsis thaliana DR5::GFP plants. Five-day-old plants were transferred to media containing the different SA treatments and plant hormones were measured 5 days later. The scale represents 100 µm.

**Figure 11 ijms-23-14019-f011:**
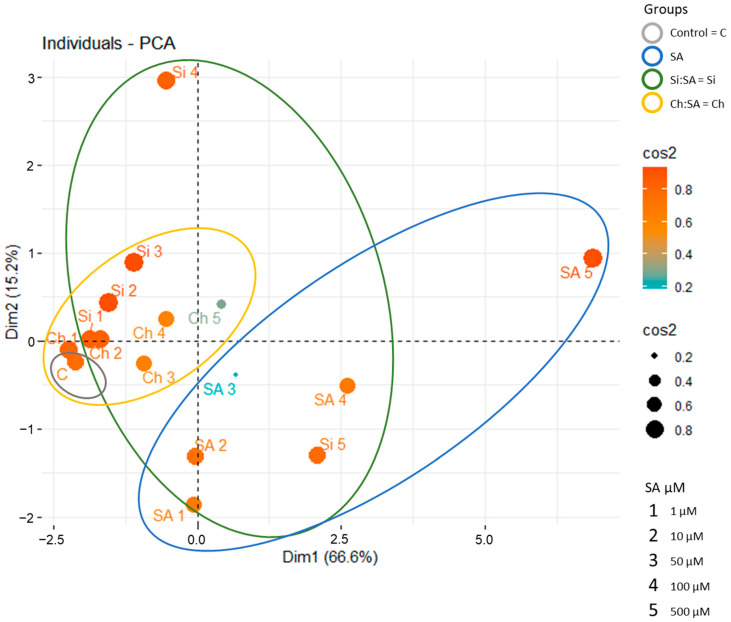
Individual-PCA of Col-0 Arabidopsis plants treated with free SA, Si:SA and Ch:SA at 1, 10, 50, 100 and 500 µM doses.

## Data Availability

Data are contained within the article.

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
