# Peer review of "Encapsulation Reduces the Deleterious Effects of Salicylic Acid Treatments on Root Growth and Gravitropic Response"

_ijms, 2022, doi:10.3390/ijms232214019_

Round 1

Reviewer 1 Report

In this manuscript, the author explains that encapsulation reduces the deleterious effects of salicylic acid treatments on root growth and gravitropic response. The effect of SA on roots and rosettes was analyzed, determining plant morphological characteristics and endogenous hormone levels. Free SA treatment affected length, growth rate, the gravitropic response of roots, and rosette size in a dose-dependent manner. This damage was due to the increase of root endogenous SA concentration that led to a reduction in auxin levels. The encapsulation process reduced the deleterious effects of free SA on root and rosette growth and in the gravitropic response. Encapsulation allowed a controlled release of the SA, reducing the amount of hormone available and the uptake by the plant, mitigating the deleterious effects of the free SA treatment.

The overall manuscript is very basic, with only a control experiment done. The study lacks molecular work. The main drawback of this study is that it only talks about encapsulation reducing the deleterious effects of salicylic acid treatments on root growth and gravitropic response, but it does not explain how? At this current stage, it cannot be accepted in IJMS.

Author Response

Thank you for your comments. We believe that the manuscript is aligned with the scope of the International Journal of Molecular Sciences, as papers studying the effects of carriers on the delivery of molecules in other biological systems have been recently published. This work studies the effect of SA on plant morphology and physiology. We have also analysed endogenous hormonal levels. We stated that the damage caused by free SA treatments was due to the increase of root endogenous SA concentration that led to a reduction in auxin levels. Therefore, we believe that the manuscript provides new insights on the hormonal mechanism of action. To clarify this point, we have modified the text to highlight the molecular mechanisms by which encapsulated SA works. We have also further explained the encapsulation process, highlighting the analogies and differences between capsules. Encapsulation allowed a controlled SA release, reducing the amount of hormone available and the uptake by the plant. We hope that the improvements made with the feedback of all reviewers will satisfy your requirements:

In section 2.1. the following paragraph has been added to clarify the encapsulation process: In this study, we found that the process differs between capsules. In the case of amorphous silica, a SA gradient from the surface to the centre of the encapsulated sample was observed, with a potential saturation of the silica porous surface depending on the Si:SA ratio. In the case of chitosan, this process is produced by entrapment of the bioactive molecule by the chitosan polymeric chains and subsequent cross-linking with TPP-Na. These differences in the encapsulation process were reflected in the SA release rates, being faster in the case of silica. However, no differences were observed in the release method, produced in both cases by breakage of the carrier matrix structure and not by diffusion of the encapsulated molecule”.

In section 2.5 the following paragraph has been added to clarify the molecular mechanism of the encapsulated SA improvement: From these data, it can be concluded that one of the improvements of the use of encapsulated SA is to prevent auxin flow interferences

In the same sense, in section 2.6 the following paragraph has been added: Therefore, the hypothesis of exogenous SA treatments modifying endogenous IAA levels is further supported, allowing us to conclude that part of the deleterious effect of exogenous SA is due to the reduction of root IAA levels and that encapsulation avoids this process

In this work, after an initial assay of the effect of SA doses on the development of Arabidopsis plants, the following experiments were performed: First, the effect of the SA treatments on root and rosette physiology and morphology was evaluated both at short and long periods. Then, the gravitropic response of treated plants was studied and, finally, DR5::GFP lines were used to evaluated auxin fluxes in roots. Moreover, in each case, the following treatments were tested: Free-SA at five different concentrations, encapsulated SA (two different capsules) at five different concentrations, both empty capsules, and controls. This has been clarified in the text of the manuscript. The paragraph, included in section 3, reads now as follows: After an initial study of the effect of SA doses on the development of Arabidopsis plants, the following experiments were performed: First, the effect of the SA treatments on root and rosette physiology and morphology was evaluated both at short and long periods. Then, the gravitropic response of treated plants was studied and, finally, DR5::GFP lines were used to evaluated auxin fluxes in roots

Reviewer 2 Report

It is known that CA is an inhibitor of ion intake, a regulator of the transport of organic substances through the phloem to the roots and gravitropism. Under the influence of CA treatment, the level of ABA increases, as well as IAA, which is known to be important in regulating the activation of metabolic processes underlying plant growth. At the same time, CA treatment prevents a sharp accumulation of ABA caused by moisture deficiency and salinization of the medium and a drop in the level of IAA and cytokinins in plants. This may, at the hormonal level, reflect the salicylic acid-induced resistance of plants to abiotic environmental factors and an increase in their productivity.

The role of CA in plant pathogenesis seems to be quite complex and contradictory. The manifestation of its protective functions depends on many factors (concentration, the interval between treatment and infection, the type of plant, etc.) and sometimes it turns into its opposite - the induction of susceptibility to the disease. 

Therefore, I would still like to know about the specific conditions of plant cultivation. Have studies been conducted under stressful conditions, as the result may differ from those obtained by the authors.

In general, the work is interesting and can be published.

Author Response

Thank you for the positive comments and the interesting information about CA that we will have under consideration for future studies. So far, we have not studied the SA precursors in our system, but it could be interesting to include the CA effect in follow-up works.

Regarding the specific conditions of plant cultivation, plants were grown under no stress conditions. We have modified the Materials and Methods section to state this information clearly. The paragraph reads now as follows: Seeds were germinated under no stress conditions and petri dishes were vertically arranged (Figure 1) in growth chambers (SANYO MLR-350, Sakata, Gunma, Japan) for 5 days under 16 h light/8 h dark cycles at 22.5°C and 60% relative humidity.”

Reviewer 3 Report

Encapsulation reduces the deleterious effects of salicylic acid treatments on root growth and gravitropic response

Among many, the fundamental role of biomolecules such as salicylic acid, jasmonic and abscisic acid is undeniable. The action of these phytohormones can effectively attenuate the abiotic stresses that weaken crops.

General comments:

The work is well structured, with very well-elaborated figures and graphics that skillfully explain the results.

Only two specific comments:

Abstract: The acronym SA appears here for the first time; I suggest you specify the full name here or even in the title.

Figure 1: section b3) please correct Rossete area with Rosette area.

Author Response

Thank you for the positive comments.

Regarding the specific comments, we have modified the manuscript to correct these mistakes. The paragraph reads now as follows:

Abstract: The role of salicylic acid (SA) on plant responses to biotic and abiotic stresses is well documented”.

Figure 1: “b3) Rosette area analysis

Reviewer 4 Report

This paper deals with the investigation of the growth traits in the Arabidopsis plants treated with free SA and encapsulated SA. The encapsulated SA treatments improved the growth inhibition, indicating that those treatments are attractive methods for the cultivation of crops in future.

I have two comments.

1) Could you briefly explain or discuss the two encapsulation methods in respect to chemical aspects?

2) The contents of JA and ABA were measured in this paper? I could not catch the descriptions of these hormones. (P7, L188; P11, L271) 

Author Response

Thank you for the positive comments.

1) Regarding the specific encapsulation methods, in previous studies (29) we found that the process differs from one capsule to another. In the case of amorphous silica, a gradient of SA from the surface to the centre of the encapsulated sample was observed, with a potential saturation of the silica porous surface depending on the Si:SA ratio. In the case of chitosan, this process is produced by entrapment of the bioactive molecule by the chitosan polymeric chains and subsequent cross-linking with TPP-Na. These differences in the encapsulation process were reflected in the SA release rates, being faster in the case of silica, but no differences were observed in the release method, produced in both cases by breakage of the carrier matrix structure and not by diffusion of the encapsulated molecule. The manuscript has been modified to include this information. The paragraph, included in section 2.1, reads now as follows: In this study, we found that the process differs between capsules. In the case of amorphous silica, a gradient of SA from the surface to the centre of the encapsulated sample was observed, with a potential saturation of the silica porous surface depending on the Si:SA ratio. In the case of chitosan, this process is produced by entrapment of the bioactive molecule by the chitosan polymeric chains and subsequent cross-linking with TPP-Na. These differences in the encapsulation process were reflected in the SA release rates, being faster in the case of silica. However, no differences were observed in the release method, produced in both cases by breakage of the carrier matrix structure and not by diffusion of the encapsulated molecule”.

2) Regarding the JA and ABA determinations, both molecules, indeed, have been determined in this work, and the results are shown in Figures S7 and S8, included with the supplementary material. The manuscript reflects this information on the current page 10, lines 267 to 269.

Round 2

Reviewer 1 Report

I am happy with the author's reply and think that manuscript can be accepted in the current format.